# Implications of Abscisic Acid in the Drought Stress Tolerance of Plants

**Shahid Ali [1], Kashif Hayat [2,3], Amjad Iqbal [4] and Linan Xie [1,5,]**

[1]  College of Life Sciences, Northeast Forestry University, Harbin 150040, China; shahidsafi926@gmail.com
[2]  Key Laboratory of Urban Agriculture, Ministry of Agriculture and Rural Areas, Shanghai Jiao Tong University, Shanghai 200240, China; khayat97@sjtu.edu.cn
[3]  School of Agriculture and Biology, Shanghai Jiao Tong University, Shanghai 200240, China
[4]  Department of Agriculture, Abdul Wali Khan University Mardan, Mardan 23200, Pakistan; amjadiqbal@awkum.edu.pk
[5]  Key Laboratory of Saline-Alkali Vegetative Ecology Restoration, Ministry of Education, College of Life Science, Northeast Forestry University, Harbin 150040, China
*  Correspondence: Linanxie@nefu.edu.cn

**Abstract:** Drought is a severe environmental constraint, which significantly affects plant growth, productivity, and quality. Plants have developed specific mechanisms that perceive the stress signals and respond to external environmental changes via different mitigation strategies. Abscisic acid (ABA), being one of the phytohormones, serves as an important signaling mediator for plants' adaptive response to a variety of environmental stresses. ABA triggers many physiological processes, including bud dormancy, seed germination, stomatal closure, and transcriptional and post-transcriptional regulation of stress-responsive gene expression. The site of its biosynthesis and action must be clarified to understand the signaling network of ABA. Various studies have documented multiple sites for ABA biosynthesis, their transporter proteins in the plasma membrane, and several components of ABA-dependent signaling pathways, suggesting that the ABA response to external stresses is a complex networking mechanism. Knowing about stress signals and responses will increase our ability to enhance crop stress tolerance through the use of various advanced techniques. This review will elaborate on the ABA biosynthesis, transportation, and signaling pathways at the molecular level in response to drought stress, which will add a new insight for future studies.

**Keywords:** ABA biosynthesis; drought stress; gene expression; signaling network; transporters

## 1. Introduction

The world population is regularly increasing and is estimated to grow by more than 2.4 billion by 2050 [1]. It is essential to increase food production and quality up to 70% by 2050 to feed the increasing population. Climatic stresses, like heat, cold, salinity, and drought have a drastic effect on plant production. Moreover, such stresses can significantly decrease crop productivity, leading to major losses worldwide [2]. Plants trigger several physiological, biochemical, and molecular responses that influence plant morphology and various cellular processes to counter environmental stresses [3]. Drought stress refers to the shortage of water precipitation that causes water deficits in soil water. Water deficiency in soil lowers the groundwater level, which may hamper the growth and survival of plants. Drought stress may lead to harmful effects in plants through the disruption of various activities, such as carbon assimilation, exchange of leaf gasses, turgor pressure, and increased oxidative damage. Likewise, drought stress may also affect ion balance, enzymatic activity, stem extension, leaf size, and root proliferation. All the above-mentioned losses due to stresses result in reduced yield [4,5].

Plants are sessile in nature and are unable to avoid both biotic and abiotic stresses. Plant species lack a well-defined immune system, but they have signaling mechanisms to ensure optimization growth and secure overall fitness. For example, plant hormones serve as central integrators to link and reprogram complex developmental mechanisms and adaptive stress processes. Abscisic acid (ABA) is an essential key regulator that regulates many aspects of plant growth and development, including embryo maturation, seed dormancy and germination, floral induction, and root growth. ABA also helps in reducing the adverse effects related to stresses in plants, such as drought. ABA enables the plant species to adapt to a continuously changing environment and maintain the physiological processes under drought stress. The plants' physiological processes can be regulated by the modulation of the expression of various ABA-responsive genes responsible for the opening and closing of the stomatal aperture [6,7].

In fact, previous researchers have identified the core components of ABA biosynthesis and its signaling pathways through genetic, molecular, biochemical, and pharmacological approaches. With the help of these approaches, scientists have identified various components involved in ABA biosynthesis and signaling pathways in *Arabidopsis* and maize during seed germination. In this review, we describe the recent signs of progress made in the studies of ABA-dependent drought tolerance in plants through stomatal regulation, root development, and seed germination.

## 2. Sensing Abiotic Stresses

Adverse environmental conditions, such as abiotic (salinity, extreme temperature, drought, nutrient deficiency, and heavy metals) and biotic stresses (pathogen infection and herbivore insects) severely affect plant growth, productivity, and food security [8,9]. Among abiotic stresses, the drought stress is characterized as a low availability of water for a prolonged period of time. In order to combat this stress, the first step taken by a plant species is the perception of stress signals by plant cells. The signals are generally associated with the multiple genes and cellular signaling pathways that further modulate molecular and physio-morphological responses [10]. Drought-stress-signaling pathways start with the perception of stress by transmembrane histidine kinase, a plasma membrane receptor. In *Arabidopsis thaliana*, the histidine kinase domain containing proteins (AtHKT1) activates the downstream signaling cascade to induce gene expression [11]. It triggers a disruption of intracellular calcium levels, causing the calcium sensors to activate a phosphorylation cascade. Calcium ($Ca^{2+}$) is recognized as an essential second messenger in the signaling pathways, that enables the plant species to adapt to changes in their surrounding environment [12]. Plants perceive the stimuli that are followed by the increase in intracellular $Ca^{2+}$ concentration. The increase in intracellular $Ca^{2+}$ concentration occurs either instantaneously or after a time lag, in single or many intracellular compartments, such as cytosol, chloroplast, mitochondria, nucleus, etc. The increasing cellular transient $Ca^{2+}$ is recognized by calcium sensors, including calmodulins (CaMs), calmodulin-like proteins (CMLs), calcineurin B-like proteins (CBLs), and calcium-dependent proteins kinases (CDPKs/CPKs). This further targets the downstream events, such as gene expression and protein phosphorylation [13,14]. CaMs, CBLs, and CMLs are classes of sensor-relay proteins, whereas CPKs work as sensor responders, possessing $Ca^{2+}$ binding and kinase activities [15,16]. CaMs are small proteins (~17 KDa) that have been well studied at the structural level. CaMs contain C- and N- terminal globular regions that are separated by a flexible central helix. The globular regions of CaMs consist in a pair of $Ca^{2+}$ binding EF-hand motifs [17,18]. CMLs are closely related to CaMs, but present in higher plants. CMLs can vary in length and contain one to six EF-hand motifs. Plant CaMs and CMLs have a functional role in the developmental process and responses to different stresses. Under drought, salinity, and cold, the expression level of CaMs and CMLs in plant species were noticed to increase significantly. Furthermore, the transgenic lines with overexpression of CaMs and CMLs exhibited greater stress tolerance as compared to the wild type [19–22], whereas mutants were more sensitive to the applied stress [23]. Moreover, there is evidence that CaMs and CMLs interact with target proteins to regulate stress adaptations either positively or negatively [24,25].

Previous studies characterized 7 CAMs and 50 CMLs in *Arabidopsis* that were bound to calmodulin-binding transcription factors (CAMTAs). The CAMTAs acted as a signal responsive protein, and responded to extreme temperature, ionic, and osmotic stress. CAMTA was first characterized in tobacco during the screening of CaM-binding proteins [26–28]. After their characterization, various plants have been reported to have a variable number of *CAMTA* genes, for example, *Nicotina tabaccum* (6) [29], Citrus (9) [30], *Medicago truncatula* (7) [31], *Arabidopsis thaliana* (6) [28]. *Lycopersicum esculantum* possesses 7 *CAMTA* genes, and these genes expressed differentially in various stress conditions [32]. CAMTA proteins own multiple predicated functional domains, such as CG-1 DNA-binding domain, TIG domain, ankyrin repeats, and CaMBD, which binds with CaM in a calcium-independent manner [33–35]. CAMTA TFs act as pivotal integral elements of calcium-mediated stresses and hormonal signaling pathways. CAMTA recognizes and directly binds to the ((A/C)CGCGC/G/T), (A/C(GTGT)) sequence of the target gene through the CG-1 domain and thereby regulates its expression [27]. In rice, the CAMTA homologs bind to cis-elements (A/C)CGTGT, which encompass the abscisic acid ABA-responsive elements [36,37]. In *Arabidopsis*, CAMTA1,2,3 activated CBF (C-repeat/DRE-binding factors) transcription factors, which contributed to freezing and low-temperature tolerance. In *Arabidopsis*, calmodulin-binding transcription activator1 (AtCAMTA1) plays a role in the expression regulation of membrane integrity genes through ABA production in response to drought stress [38]. The AtCAMTA1 is also involved in the auxin signaling responses; the exogenous auxin significantly changes its expression pattern in a cell-specific manner [39]. In wheat, TaCAMATA4 negatively regulates the defense response against pathogens [40], whereas in *Arabidopsis*, AtCAMATA3 [41] and AtCAMTA5 [42] play a role in salt tolerance.

CBL proteins consist of four EF-hand domains, which capture intracellular $Ca^{2+}$ [43,44], without enzymatic activity. However, upon $Ca^{2+}$ binding, these proteins interact with a kinase known as CBL-Interacting Protein Kinase (CIPK). CIPK is similar to the AMP-dependent Kinase (AMPK) from animals and sucrose nonfermenting (SNF) protein kinase from yeast in the kinase domain [45,46]. In plants, the SNF1-related kinase (SnRK) is composed into three subfamilies: (a) SnRK1, with a role in carbon and nitrogen metabolism; (b) SnRK2; and (c) SnRK3, with a vital role in the regulation of stress signaling [47]. The CIPK protein, also known as SnRK3, contains a Ser/Thr protein kinase domain. Besides, CIPKs have a conserved NAF/FISL motif in the C-terminus regulatory domain and the N-terminal kinase domain [48]. Moreover, CIPKs also contain a protein phosphatase interaction (PPI) domain, which interacts with the phosphatase (PP2C) protein [49]. The available data suggested that calcium mediates the formation of a stable CIPK-CBL complex, which functions together in decoding calcium signals initiated by different environmental stimuli. In *Arabidopsis*, 10 CBLs and 26 CIPKs were characterized to date. Various techniques have confirmed the upregulation of CBL-CIPK proteins during osmotic stress, salinity, and the response to ABA. This suggests that CBL-CIPK proteins have a role in ABA signaling and salinity tolerance. During screening of the salt overly sensitive (SOS) phenotypes in *Arabidopsis*, a CBL-CIPK pathway was shown to be comprised of AtCBL4 (SOS3), AtCIPK24 (SOS2), and plant membrane-localized $N^+/H^+$ antiporter (SOS1) [50]. The CBL-CIPK pathway was found to be involved in regulating sodium ($Na^+$), magnesium ($Mg^{2+}$), potassium ($K^+$), and nitrate ($NO^{3-}$) transport across the plasma membrane or vacuolar membrane [45,51–53]. The CBL10-AtCIPK24 complex induced the SOS1 on the tonoplast in different salt-tolerance pathways to confer salt tolerance. The CBL1/CBL2-CIPK23 complex in the plasma membrane modulates the *Arabidopsis* $K^+$ Transporter1 (AKT1) to regulate the $K^+$ content in root and guard cells [54,55]. The AtCBL1-AtCIPK7 kinase complex has a significant role in cold tolerance as well. In the transgenic plant, the over-expression of AtCBL5 increases osmotic and salt tolerance. The multivalent interaction of the CBL2/3 and CIP/26 complex decreases the toxicity of $Mg^{2+}$ and sequesters it in the vacuole [51]. In *Arabidopsis*, the CIPK3 expression was noted to change in response to ABA, salt, and high cold stress conditions. Moreover, CIPK3 mediates the interaction of abiotic stressors and ABA through ABA-dependent and independent pathways [56,57].

CPKs are the third component of calcium-sensing complexes in plants. Furthermore, they are sensor responders with the ability to self-modify the confirmation through enzymatic action. This shows that CPKs have a role in calcium-sensing, which then responds to the downstream phosphorylation in order to encode the stress cues [58–60]. CPKs are monomeric, highly conserved proteins that are involved in plant growth and development, defense, stress signaling, and proteasome regulation [61,62]. The structure of CPKs consists of five domains, i.e., the regulatory, autoinhibitory, kinase, C-terminal, and N-terminus domains. CPK proteins have potential N-myristoylation and palmitoylation motifs to make an association with the membrane in the N-terminal domain, but some CPKs are also located in the cytosol [62–64]. It was shown that CPKs positively or negatively regulate salinity and drought stresses [65]. In *Brassica napus*, 25 CPKs were identified, with a suggestion that BnCPK4 interacts with Protein Phosphatase 2C (PP2C) to regulate ABA-responsive transcription factors, such as ABF1 andABF4, during drought stress [66]. The *Arabidopsis* genome encodes 34 CPKs, that are divided into AtCPK3, 6, 10, and 32. These CPKs are responsible for the control of ABA and $Ca^{2+}$ signaling pathways in order to regulate the stomatal movements under drought stress [67–69]. The overexpression of CPK4 enhanced ABA sensitivity during seed germination and seedling growth [70]. AtCPK4, 11, and 32 have an essential role in the regulation of ABA signaling pathways through phosphorylation of two ABA-responsive transcription factors (ABF1 and ABF4) under stress conditions [71,72].

## 3. ABA Biosynthesis

ABA is a small sesquiterpene-derived carotenoid, and its biosynthetic and signaling pathways have been widely studied at almost all enzymatic steps through molecular-genetic, biochemical, and pharmacological approaches [73]. In addition to plants, ABA is also synthesized by plant pathogenic fungi, using two distinct pathways. The direct pathway operates by phytopathogenic fungi through mevalonate pathways in which the intermediate has no more than 15 carbon atoms [74,75]. Plants use the carotenoid pathway to synthesize ABA, which is known to be the indirect pathway. ABA biosynthesis starts from zeaxanthin, a specific carotenoid, and the zeaxanthin epoxidase (ZEP) converts zeaxanthin into violaxanthin via antheraxanthin into two successive epoxidations [76]. The former requires the *Arabidopsis* ABA4 gene, which encodes for neoxanthin synthase and an unknown isomerase to convert all-trans-violaxanthin to 9′-cis-neoxanthin through an all-tans-neoxanthin pathway. The latter converts all-trans-violaxanthin to 9′-cis-violaxanthin directly by an unknown isomerase [77]. Both 9-cis-xanthin and 9-cis-violaxanthin served as a substrate for xanthoxin, which acts as a growth inhibitor [78]. The reaction is catalyzed by 9-cis-epoxycartenoid dioxygenase (NCEDs) encoded by *VIVPAROUS14 (VIP14)* in maize. The produced xanthoxin is then relocalized from plastid to cytoplasm. In the cytoplasm, xanthoxin is converted through the enzymatic action into abscisic aldehyde by ABA-dificient2 (ABA2), which belongs to the short-chain dehydrogenase/reductase (SDR) family. In the last step, abscisic aldehyde oxidase (AAO) converts the abscisic aldehyde to abscisic acid (Figure 1) [79]. AAO3 proteins consist of molybdenum (Mo) as a cofactor, which is activated by MoCosulfurase.

Different enzymes play an important role in the ABA biosynthesis pathway. The NCEDs function as the key rate-limiting factors, where NCED3 is the essential enzyme for ABA synthesis [80]. In *Arabidopsis*, five *NCED* genes (AtNCED2, AtNCED3, AtNCED5, AtNCED6, AtNCED9) are identified, which are localized in plastids and have an active role in ABA synthesis [81]. The gene expression analysis in *Arabidopsis* showed that NCED genes express in different regions of the plants; for example, *NCED2* and *NCED3* are highly expressed in the roots, whereas the expression of *NCED5, NCED6,* and *NCED9* is induced during late maturation [82].

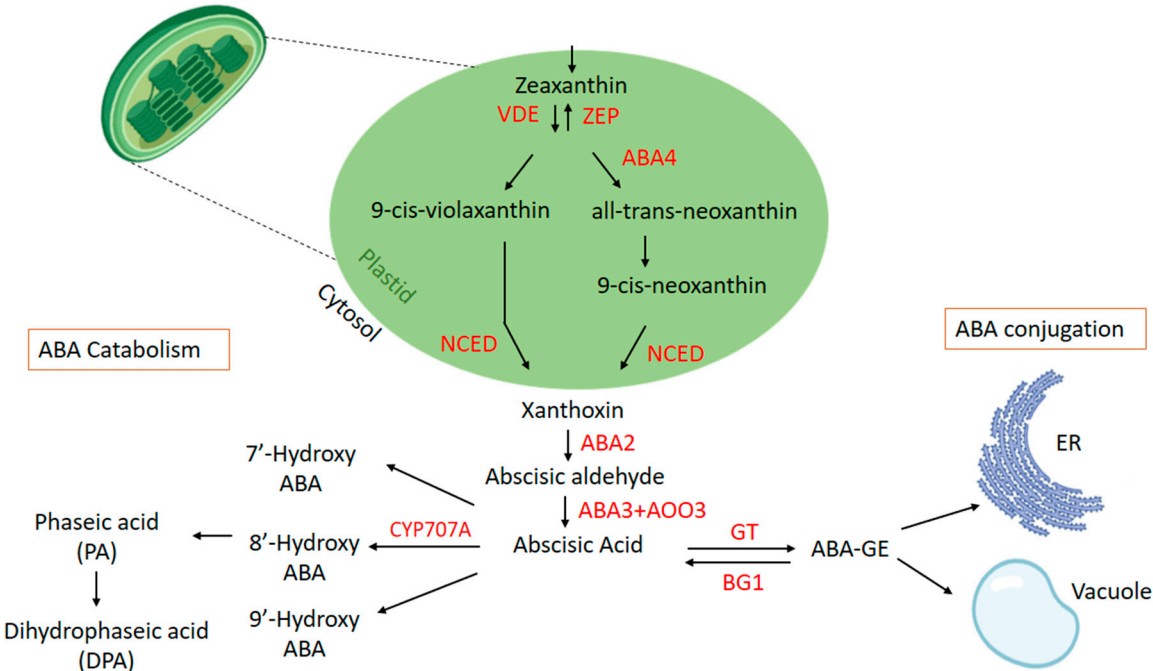

**Figure 1.** Scheme of abscisic acid (ABA) biosynthesis, catabolism, and conjugation pathways. ABA is synthesized form β-carotene precursors through oxidative reactions in the plastids and catalyzed by several enzymes (ZEP, VDE, ABA4, NCED) that convert zeaxanthin to xanthioxin. The cytosol xanthioxin was converted into ABA employing an oxidation reaction that is mediated by three enzymes (ABA2, AAO3, and ABA3). In the catabolic pathway, the ABA is converted to PA and DPA, which is mainly catalyzed by CYP707A. In the ABA conjugation, ABA is glycosylated into ABA-GE by ABA glycosyltransferase (GT), that can be transformed into the endoplasmic reticulum (ER) and vacuoles. By glycosyl hydrolase BG1, the ABA-GT can be converted back to free ABA. Abbreviation: ZEP, zeaxanthin epoxidase; VDE, violaxanthin de-epoxidase; ABA4, ABA-deficient4; NCED, nine-cis-apoxicarotenoid dioxygenase; ABA2, ABA-deficient2; AAO3, abscisic aldehyde oxidase, ABA3, ABA-deficient3.

## 4. Regulation of ABA Biosynthesis through Abiotic Stress

ABA plays a vital role in response to abiotic stress, especially in water-deficient environments, where plant species elevate the cellular level of ABA to combat drought stress [83]. Under drought stress, *NCED3* is a dominant contributor to ABA production in *Arabidopsis*. In plants, to cope with the water deficit conditions, ABA acts as a central regulatory component that regulates the stomatal movement [84,85]. *NCED* genes were noticed to be upregulated in drought stress conditions in *Arabidopsis* [86], tomato [87], maize [88], avocado [89], cowpea [90], and bean [91]. Previous studies demonstrated that sulfate acts as a chemical signal to drought and induce NCED3, which increases the ABA level and promotes stomatal closure [92]. Recently, *NCED3* expression in leaves has been reported to be regulated by root-derived peptide CLE25 (clavata3/ESR-Related 25), which transmits the water-deficiency signals. Moreover, its transcripts are significantly induced in the vascular tissue of the roots when plants are exposed to drought stress. The root-derived CLE25 levels increase in the leaves in case of drought, which is further associated with receptor-like proteins (RLKs), BAM1 (Barley Any Meristem1), and BAM3 stress [93].

Furthermore, class II HD-ZIP transcription factors HAT1 and HAT3 act as ABA regulators. *HAT1* overexpression exhibits less tolerance to water stress, while a double mutant of *hat1* and *hat3* significantly enhances tolerance to drought stress. It was confirmed that HAT1 binds to the promoter region of *NCED3* and negatively regulates its expression [93]. Moreover, in *Arabidopsis*, other transcription factors [ANAC2 (*Arabidopsis* NAC domain-containing protein 2), BDG1 (9-cis-epoxycaroteniod dioxygenase

defective 1) and WRKY57 (WRKY DNA-binding proteins 57)] modulate *NCED3* expression [94–96]. These results confirm that NCEDs are a crucial enzyme in ABA synthesis and essential for coping with the multiple stress conditions.

## 5. ABA Catabolism and Conjugation

When plants are exposed to different environmental conditions, the level of ABA significantly increases. Therefore, the maintenance of a basal ABA level is paramount for appropriate growth and development. Catabolism of ABA is one of the determinants of the endogenous levels of ABA, affecting many aspects of growth and adaptive abiotic stress responses. Therefore, the level of ABA is controlled by both catalytic hydroxylation and ABA conjugation. Hydroxylation pathway triggers the degradation of ABA at three various methyl groups (C-7′, C-8′, C-9′). C-8′ is the predominant position for the hydroxylation reaction to form 8′-hydroxy ABA (8′OH-ABA). However, this is unstable, spontaneously isomerizes to phaseic acid (PA), and finally reduces to dihydrophaseic acid (DPA) [97,98]. The ABA C-8′ hydroxylation reaction is mediated by the protein encoded by the CytochromeP450 (CYP)707A subfamily mono-oxygenase (CYP797A1-4) [99]. In *Arabidopsis*, VIP1 (VirE2-interacting protein1) and ZIP transcription factors bind to the promoter regions of CYP707A1 and CYP707A3 to control ABA content during rehydration [100]. Previous studies also showed that in the VEGETATIVE PHASE (SVP), a MADS-box transcription factor binds to the CArG motifs in the *CYP707A1,3* and *AtBG1* promoter. This further decreases the expression of *CYP707A1,3* and elevates the expression of *AtBG1* in *Arabidopsis* leaves to enhance the ABA contents [101].

Besides the de novo synthesis, the concentration of ABA is also enhanced by the hydrolysis of glucosyl ester (ABA-GE), a glucose conjugated form of ABA. The production of ABA from ABA-GE is a single-step process that contributes to increase of the ABA concentration, which is required for the physiological needs of plants. In cytosol, uridine diphosphate glucosyltransferase (UGTs) and UGT71B6/7/8 enzymes catalyze the conversion of the ABA to an inactive form of ABA-GE that accumulates in the vacuole and endoplasmic reticulum [102]. It has been documented that in *Arabidopsis* leaves, ABA-GE is deliberately imported into the endoplasmic reticulum via unidentified transporters, in response to dehydration [103,104]. Additionally, the UGT71Bs are important for ABA homeostasis, which regulates plant growth and various environmental stresses, such as salinity, drought, etc. [105].

## 6. ABA Signaling Pathway

When plants are exposed to different environmental stresses, like drought, high and low temperature, or high salinity, their growth is modulated by the coordination between several plant hormones, proteins, and regulatory factors. Among the phytohormones, ABA plays a key role in inducing various responses, such as stomatal closure and the expression of stress-responsive genes. The ABA response to drought stress can be slow or rapid. The slow response includes the expression of target genes [106], while a rapid response is mediated by the gating of ion channels [107,108]. The ABA signaling cascade consists of a three-step regulatory process, including the receptor, protein kinase mediator, and targets. The perception of ABA through ABA receptors initiated the ABA-mediated signaling cascade. Several types of ABA receptors have been reported, but their exact role and nature have not been confirmed. The major component of ABA perception and signaling consists of soluble cytoplasmic PYrabactin Resistance (PYR)/PYrabactin Resistance like the (PYL)/Regulatory Component of the ABA Receptor (RCAR) family of START proteins. The group A PP2Cs (Protein Phosphatase 2C) consists of the negative regulators and subfamily of the SNF1-related kinases (SnRK2), which are positive regulators [109–111].

The ABA-bound receptors allow for the sequestration of PP2C, which is the negative regulator of the ABA signaling pathway. There are nine PP2Cs, but the best-studied PP2Cs in ABA signaling pathways include HAB1 (Homologue of ABA-Insensitive1), HAB2, ABI1 (ABA-Insensitive1), ABI2, and PP2CA [112–114]. In the absence of ABA, PP2C physically interacts with SnRK2 (Sucrose non-fermenting Related Kinase2) to be inactivated through the dephosphorylation of its kinase activity

loops [115–117]. In the presence of ABA, the receptors PYR/PYL/RCAR5 form a complex with PP2C, which dissociates PP2C from SnRK2 and allows it to activate by autophosphorylation (Figure 2) [118]. It has been documented that in the absence of ABA, several PYLs also interact to inhibit the PP2C, but the inhibition is weaker than the ABA-bound PYLs [119]. The active protein kinase (SnRK2) then activates the downstream signaling effectors (ABF transcription factors). This modulates the stress-responsive genes' expression [118], or gene encoding ion channels, such as SLAC1 (slow anion channel associated1), SLAH3 (SLAC1 homolog protein3) and QUAC1 (quick activating anion channel1), thereby regulating the stomatal movements [115,116,120,121].

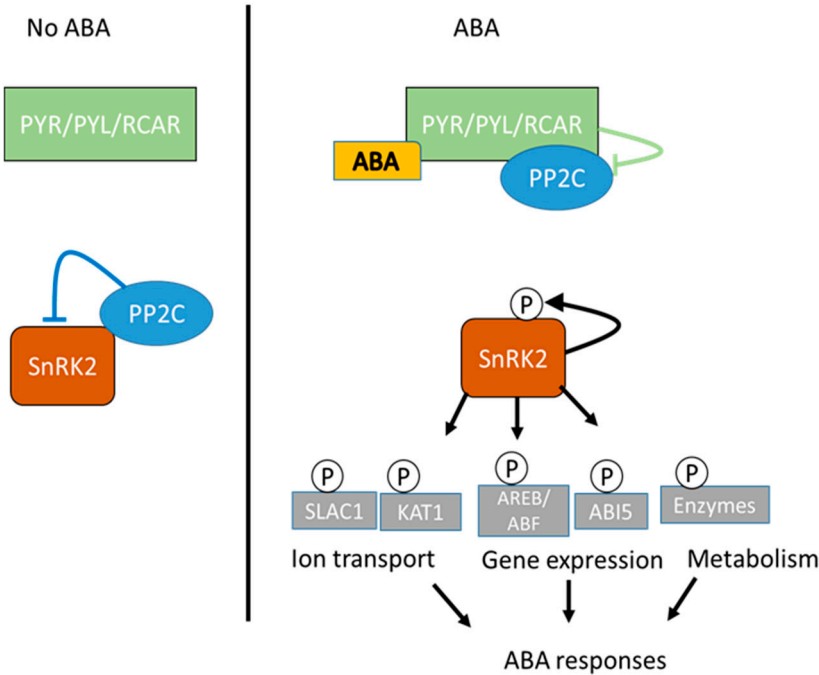

**Figure 2.** ABA-signaling pathway in the plant without ABA, and the presence of ABA. The ABA signaling pathway consists of three core components, including PYR/PYL/RCAR5 ABA receptors, a PP2C negative regulator, and a SnRK2 positive regulator. In the absence of ABA, PP2C negatively regulates SnRK2 by direct association and dephosphorylation, so that SnRK2 is not able to phosphorylate the downstream substrate. In the presence of ABA, the PYR/PYL/RCAR5 receptors bind to ABA and interact with PP2C to inhibit phosphatase activity and release the snRK2s. In turn, SnRK2 is activated by autophosphorylation of the activation loops and phosphorylate downstream factors, including AREB/ABF transcription factors and other factors such as proteins involved in ions channels, etc.

Many drought-responsive genes are involved in ABA-signaling pathways, although several drought-induced genes do not respond to ABA-signaling, showing that ABA-independent signaling pathways also regulate the response to drought stress. Heat, drought, and high salinity stresses induce the dehydration-responsive element-binding (DREB) proteins, which are the members of AP2/ERF family transcription factors [122]. In *Arabidopsis*, there are eight DREBs, but the DREB2A and DREB2B are highly induced by extreme salinity, heat, and drought stress. DREB2A plays a pivotal role in inducing many genes encoding proteins that increase plants' tolerance against abiotic stress [123,124].

## 7. ABA Transports between Vascular and Guard Cells

In drought stress ABA-induced stomatal closing, it is suspected that the root is the site where drought stress signals can be perceived and ABA synthesis take place, during drought stress. Once the ABA forms, it moves to the shoot through the vasculature. However, it has recently been suggested

that the drought-induced ABA synthesis takes place in shoot vasculature and guard cells under low humidity.

A small proportion of ABA is also diffuse from the apoplast under normal conditions, but under drought stress, the apoplastic pH increases, and free ABA is unable to diffuse from the plasma membrane. Therefore, it is important to transfer ABA across the plasma membrane from the site of its synthesis to the site of action under water deficit conditions. ATP-binding cassette (ABC) proteins are abundantly distributed in the plasma membrane of the plants. In *Arabidopsis*, there are 129 ABC transporter proteins present that are distributed into 9–13 subfamilies [125,126]. These proteins function as plasma membrane-localized transporters to import/export different molecules, such as hormones, sugars, lipids, and metals across the membrane. The ABC transporter contains two transmembrane domains to recognize the substrate for translocation, and two nucleotide-binding domains responsible for the hydrolysis of ATP [127]. In *Arabidopsis'* ABC transporters, subfamily G contains over 40 members, which involve several membranes of ABA transportations.

Many members of the ABCG subfamily have been suggested to function in ABA transport. ABCG25 is a plasma membrane-localized protein responsible for the closure of vascular veins in leaves and vascular tissue of the roots. In shoots, the expression of *AtABCG25* can be induced by exogenous ABA treatments in *Arabidopsis*. Therefore, its expression strongly overlaps with the ABA biosynthesis enzyme [128]. ABCG25 can act as an ABA exporter from vascular tissue and expresses in the companion cells of phloem. The exporting activity is confirmed in nine *ABCG25*-expressing *Spodoptera frugiperda* culture cells [129]. ABCG40 is a full-size transporter that uptakes ABA and transports it in guard cells. In yeast and tobacco cells, the expression of *AtABCG40* showed a high affinity and stereospecific ABA uptake activity. The mutant with loss of ABCG40 function exhibited delayed and reduced stomatal closure in response to osmotic stress and exogenous ABA [130]. ABCG25 and ABCG40 both function as ABA intercellular distributors and transporters between vascular and guard cells, representing a stress signal within plants [131]. In addition, a half-size ABCG transporter (ABCG22) is active in drought stress tolerance, but further investigation is needed to confirm the direct uptake of ABA by the ABCG22 transporter [132]. Besides ABCG25 and ABCG40, two other *Arabidopsis* ABCG-type transporters function to deliver ABA from endosperm to the embryo in mature seeds. Expression profiles and a transport activities assay supported that AtABCG25 and AtABCG31 are localized in the endosperm and transport ABA from the endosperm to the embryo, while AtABCG30 and AtABCG40 import ABA into the embryo [133].

Furthermore, functional screening using a yeast system identified four members of NITRATE TRANPORTER1/PEPTIDE TRANPOSTER FAMILY (NPF) members that work as ABA importers. NPF4.6 (original name AIT1) is localized in the plasma membrane, suggesting that this protein mediates the ABA uptake at the site of its biosynthesis [134]. The overexpression of *NPF4.6* is characterized by low temperature in the leaf surface, possibly due to defective stomatal closure. Interestingly, *AtABCG40* and *NPF4.6* use heterologous systems to mediate cellular ABA uptake, but their expression patterns differ. *AtABCG40* is expressed in the guard cells, whereas NPF4.6 is expressed in vascular cells. NPF4.6, in association with AtABCG25, regulates ABA transport from the vascular tissue toward the guard cells [134]. In yeast, the functional screen of ABA transport recognized NPF4.6/AIT1/NRT1.2, NPF4.5/AIT2, NPF4.1/AIT3, and NPF4.2 AIT4 (Figure 3) [134] as the transporters. NPF4.6 is localized around the vascular tissue, and its mutant exhibited lower surface temperature, confirming that it has a role in ABA transportation. Chiba and his co-workers [135] used an ABA-dependent two-hybrid system and screened 45 out of 53 NPF-members in *Arabidopsis*. They have documented NPF4.1, NPF4.5, NPF4.6, and some other members, for example, NPF1.1, NPF2.5, NPF5.1, NPF5.2, NPF5.3, NPF5.7, and NPF8.3 as ABA influx transporters. Furthermore, NPF3.1 is expressed in oocytes and can accumulate ABA [136].

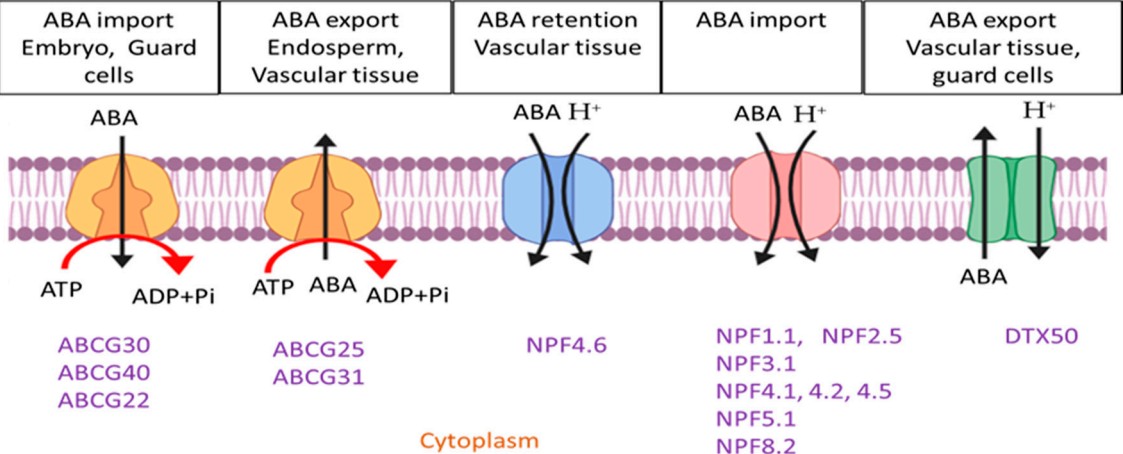

**Figure 3.** ABA transporting in plant cells. Members of the ATP-binding cassette (ABCG) family, NTR1/PTR family (NPF), and DTX50 are localized in the plasma membranes. These transporters either export the ABA from the source of its synthesis to the target site or import it from the source to the sink.

Zhang et al. [137] identified DTX50 as a new ABA transporter (ATDTX50) in *Arabidopsis*. ATDTX50 is a member of the multidrug and toxin efflux (MATE) family, that is localized in the plasma membrane and mediates the ABA efflux from the cytosol of vascular and guard cells. The reverse genetic approach in the MATE transporter showed that *dxt50* mutants can exhibit growth defects, possibly due to increasing ABA accumulation.

## 8. Role of Abscisic Acid in Improving Drought Stress Tolerance in Plants

### 8.1. ABA and Stomatal Closure

Plants absorb water from the soil by the root system and transfer it to the aerial part of the plant, where transpiration takes place. When plants are exposed to drought stress, they respond in various ways to alleviate the harsh environment. Under drought stress, the stomatal movement is an important feature to regulate the water status in the plant, which is regulated by ABA [138]. Light-induced stomatal opening occurs through inward-rectifying $K^+$ channels, such as KAT1, KAT2, AKT1, and AKT2 [139–142]. The regulation of stomatal closure is important in water conservation and pathogen defense, because it is considered as an entry point for pathogens, including viruses, bacteria, and fungi. ABA initiates the plant's defense by regulating the stomatal aperture and enhances the expression of defense-related genes against pathogen attack [143]. ABA plays a significant function in controlling the transpiration water loss of plants by manipulating guard cells' ion fluxes of stomata, both in $Ca^+$-independent and $Ca^+$-dependent pathways. Drought stress increases the level of ABA, which further enhances the cytosolic $Ca^{2+}$ in the guard cell by allowing the $Ca^{2+}$ channels to mediate the closure of stomatal pores. The high level of cytosolic $Ca^{2+}$ induces the reactive oxygen species (ROS), such as $H_2O_2$ and inositol-1-4-5 triphosphate (IP3) [144–146]. The stress signal increases the level of ABA, which induces the cellular $Ca^{2+}$ signal in plants; these signals are received downstream by calcium B-like proteins (CBLs)/CBL-interacting proteins kinase (CIPKs). The calcium sensor CBL-CIPK modulates a variety of downstream targets/ion channels. These anion channels are intensively present in the guard cells to adjust the cell volume in response to frequent environmental changes. In drought stress, they drive anions either to the inside or the outside of the cell or vacuole, which triggers the turgor pressure and reduces the guard cell volume. Two types of anion channels are present in the plasma membrane of guard cells: S(slow)-type and R(rapid)-type [147,148]. The plasma membrane S-type anion channel (SLAC1) is required for the efflux of $Cl^-$ and $NO^{3-}$ ions from the guard cell. In the inactive state, the phenylalanine residue at position 450 (Phe450) blocks the channel pore. At the same time SLAC1 is activated by kinases and the conformational changes allow for the

removal of the phe450 residue. Some kinases, for example, SnRK2.6 (also known as OST1 (OPEN STOMATA1)), CPKs/CDPKs (CPK3,6,21,23), CBL5/CIPK11, and CBL1/9-CIPK23 can activate SLAC1. Further, all these kinases activate their homologous (SLAH3), except for OST1 [67,120,149]. In the *salc1* mutant, the stomata are less sensitive to environmental cues, such as $CO_2$, ABA, and $H_2O_2$. It is also reported that (OST1)/SnRK2 proteins have a critical regulating role in ABA signaling pathways. Under water deficit conditions, the *ost1* mutant displays stomatal closure defects. However, the double mutants *ost1/snrk2.6* affect stomatal closure under normal and stress-driven ABA signaling conditions. Under an optimal ABA level, the OST1 is suppressed by PP2CA (ABI1) that inhibit the SLAC1 and the activity of SLAH3 [150–152]. ABA activates SnRK2.6/OST1 by inducing the ABI1 and PYL/PYL/RCAR protein complex formation, that allow the OST1 to phosphorylate the Ser120 residue of SLACH1 (Figure 4) [115,153]. The R-type anion channel [quick anion channel1/aluminum-activated anion channel 12 (QUAC1/ALMT12)] promotes the efflux of malate, nitrate, and chloride, which can be activated by OST1 in the guard cells. OST1 negatively regulates the bHLH transcription factors and the expression of KAT1 by blocking the ABA-responsive kinase substrate (AKS1). The AKS1 directly binds to the promoter region of KAT1 and enhances its expression. NRGA1, a putative mitochondrial pyruvate carrier, can negatively regulate the inhibition of inward K+ flow in the guard cell. The GUARD CELL HYDROGEN RESISTANT1 (GHR1) is a leucine-rich repeat localized in the plasma membrane and specifically involved in the ABA signaling pathway. The GHR1 directly interacts with the phosphorylation and regulates the SLAC1 channels [154]. ABA activates the OST1 that is capable of promoting the production of ROS in the guard cells by phosphorylating NADPH oxidase RbohD and RbohF [155,156]. ROS inhibit the ABI2, and not the ABI1, showing a ROS-mediated indirect enhancement of GHR1 activity [121,157]. Furthermore, ROS accumulation ($H_2O_2$) enhances the nitric oxide (NO) synthesis through NR1 (nitrate reductase1) activity [158,159]. NO regulates stomatal closure by both negative and positive feedbacks and attenuates RbhoD and OST1 activity, by using S-nitrosylation to trigger the degradation of ABA receptors by tyrosine nitration [160–162]. Phosphatidic acid (PA) production can be induced by NO through the activation of phospholipase C and D, which in turn activates RbhoD/RbhoF and inhibits ABI1 activities [163,164].

ABA increases the cytosolic $Ca^{2+}$ level by $Ca^{2+}$ influx into the cell via inward $Ca^{2+}$ transporters, and releases $Ca^{2+}$ from intracellular storage organelles [165–167]. ROS also activate the $Ca^{2+}$ permeable Ica channels in the plasma membrane [145,168,169]. High levels of $Ca^{2+}$ activate CBL1 and CBL9, which form a complex with CIP26 and phosphorylate RbohF [170–172]. $Ca^{2+}$ also activates other kinases, such as CPK4, 5, 6 and 11, which phosphorylate RbhoD [58,173]. Different kinases, for example, CPK3, 6, 21, and 23, as well as CIPK11(with CBL5), and CIPK23 (with CBL1 or CBL9) activate SLAC1 [116,120,173,174]. These kinases can also be inhibited by ABI1 and ABI2, like OST1. The *cpk3cpk6* double mutant or *cpk5cpk6cpk11cpk23* quadruple mutant disturbed the ABA-induced activation of an S-type channel in the guard cells [67,175]. In *cipk23* and *cbl1cbl9*, mutant opposite stomatal behaviors have been noticed after reducing leaf water transpiration and enhanced drought tolerance [54,67,173,175]. Increased osmotic stress tolerance was observed in the *cpk21* mutant without disturbing stomatal conductance, and even CPK21 was shown to activate GORK activity and an outward $K^+$ channel in the guard cell, which works synergistically with SLAC1 in stomatal closure [176,177]. Furthermore, in the *cpk21* mutant, the CPK23 gene expression was upregulated; these somewhat confusing results can be explained by the functional overlapping and compensation of CPKs [96]. CIPK23 kinase is known to activate various channels, including inward K+ channels AKT [54,178,179], but other than SLAC1, so it was speculated that this kinase worked as a negative regulator of ABA-signaling in guard cells [141,180].

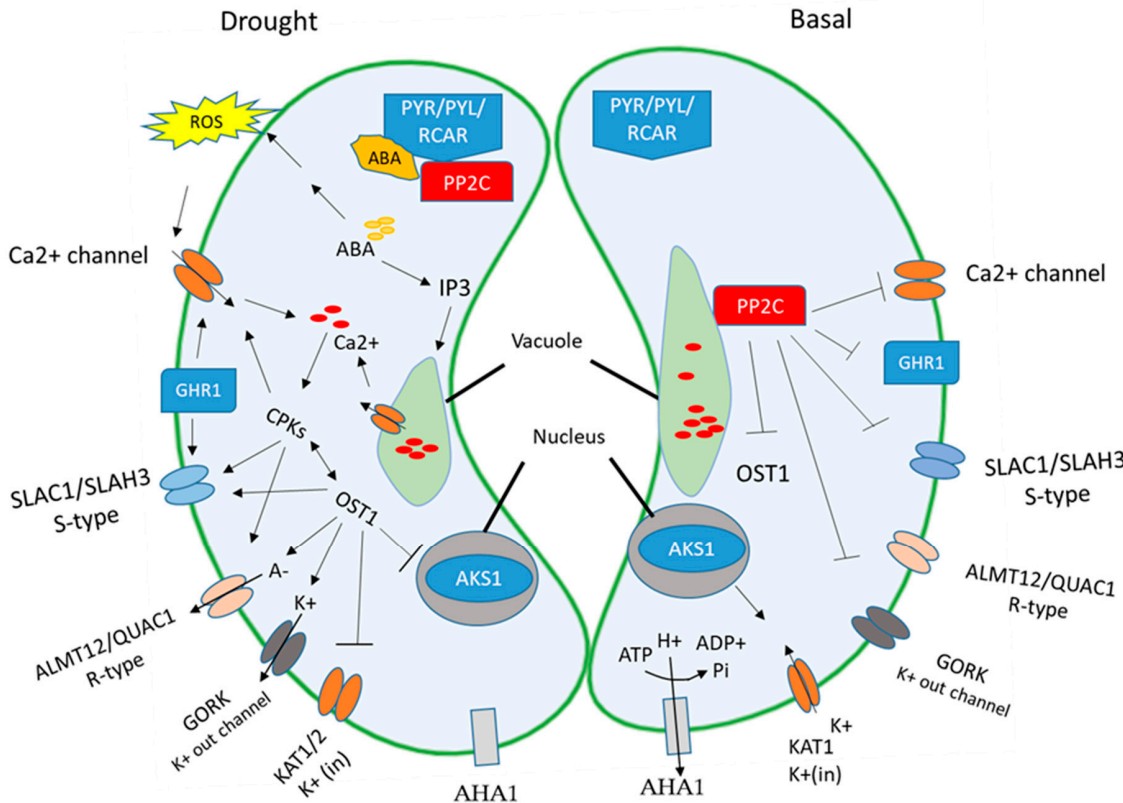

**Figure 4.** Group A PP2Cs dephosphorylate SnRK2 and S-type anion channels (SALC1), in the absence of ABA. In the activation of the H-ATPase, which leads to hyperpolarization of the plasma membrane and uptake of K through inward-rectifying channels such as KAT1 (K$^+$ channel *Arabidopsis thaliana*1), the higher concentration of K induces an inward flow on the water into the guard cells, which leads to swelling and opening of the stomata. In the presence of ABA, ABA binds to the PYR/PYL/RCAR receptor, which interacts and inhibits the PP2C by changing its confirmation, and leading to the activation of the SnRK2 protein kinase OST1. OST1 can phosphorylate ion channels, including R-type anion channel AtALMT12/QUAC1, SLAC1, and KAT1 or KUP6/8, the NAFPH oxidase RBOHD/F, and ASK family transcription factors. ABA-induced reactive oxygen species (ROS) production is mediated by NADPH oxidase. ROS possibly activate GHR1 and mediate the ABA-activation of a hyperpolarization-dependent Ca$^{2+}$ permeable cation channel and S-type anion channel. The increase of Ca$^{2+}$ cyt activates calcium-dependent kinases (CDPKs and CPKs), which phosphorylate SLAC1. These anion channels increased plasma membrane depolarization, which activates guard cell outward-rectifying K$^+$ (GORK) channels. This ion efflux leads to loss of water from guard cells, shrinkage, and hence the closure of stomatal pores.

## 8.2. Role of ABA in Root Growth and Development under Drought Stress

Plants change the growth of organs and adjust their morphogenesis in response to endogenous and environmental factors. The alternation in cell growth occurs to modify organ growth according to environmental cues, but the molecular mechanism underlying this process remains unclear. Abscisic acid regulates the dormancy and germination of the seed, inhibiting root growth in response to salinity and drought stresses [181–183]. ABA is perceived by cytosolic receptors PYR/PYL/RCAR, but in the absence of ABA, the negative regulator PP2C dephosphorylates SnRK2, thereby reducing the function of the SnRK2 kinase and inhibiting the expression of ABA-inducing genes. In the presence of high levels of ABA, the PYR/PYL/RCAR receptors mediate the PP2C inactivation and release of SnRK2 from PP2C. SnRK2 then activates several transcription factors, initiating ABA-induced gene expression [118,184,185].



Exogenous ABA causes lateral root growth quiescence, which is sustained by the knockout of ABA receptor *PYL8*. The knockout of *PYL8* and *PYL9* reduces the plant sensitivity to ABA regarding the primary root growth and lateral root growth, and a longer ABA-induced quiescence in lateral root growth. In addition, PYL8 interacts with MYB DOMAIN PROTEIN77 (MYB77) and functions in auxin-mediated lateral root growth [186]. The exogenous IAA reversed the prolonged quiescent period of the double mutant of PYL8-1, PYL9. It was also documented that PYL9 interacts with MYB44 and MYB77 in vivo. Therefore, the recovery of a lateral root from ABA inhibition is due to PYL8 and PYL9 when interacting with MYB transcription factors [187].

ABA has some negative and positive effects on primary root growth, depending on their concentration, developmental context, environmental conditions, and plant species. High levels of ABA inhibit root formation, while a low concentration stimulates it [188–190]. Low levels of ABA promoting the primary root development by maintaining the activity of the quiescent center (QC), enhancing the stem cells' activity, and suppressing the stem cells and their daughter cell differentiation in the root meristem [191,192]. A low level of ABA positively modulates the auxin signaling and transport, which stimulate root growth [189,193]. High levels of ABA inhibit root growth, whereas ethylene, auxin, $Ca^{2+}$, and ROS mediate this process [194–198], but the function of these components is still unclear.

ABA integrates with other phytohormones in order to regulate many developmental processes. The auxin gradient at the root tip determines the root's architecture [199,200]. Moreover, the auxin maxima in the stem cell niche controls the arrangement and fate of the apical meristem cells. The auxin gradient can certainly be formed by auxin influx transporter AUX1/LAX (auxin resistant1/like AUX1) and efflux transporter PINs (PIN-formed). Auxin suppresses the expression of *WOX5 (WUSCHEL RELATED HOMEOBOX5),* which is an important regulator in root development [199,200]. ABA arrested the root growth in *Arabidopsis* by reducing the auxin level. The high concentration of ABA decreased the *PIN1,3,4,7* and *AUX1* gene expression. Additionally, the mutants of *aux1* and *pin1* minimized the ABA inhibition sensitivity in the primary root growth as compared to the wild type. In *Arabidopsis*, the ARF2 (auxin response factor2) transcription factor negatively modulated the ABA-inhibited root elongation by suppressing the HB33 (Homeobox protein33) [201]. AP2 (APETALA2) domain transcription factors PLTs (PLETHs) is present in high concentration near the QC, and responsible for the regulation and maintenance of stem cell activity. A low level of PLTs stimulates cell differentiation, whereas a medium level of PLTs promotes cell division. The transcriptional gradient of PLT1 and PLT2 is highly correlated with the auxin gradient in the root meristem [200,202]. ARF2 stimulates the PLT1 expression, while decreasing PLT2 expression in ABA signaling. ABA reduces the expression of PLT1 and PLT2 proteins in the root by inhibiting the cell division in the root tips [198,203].

It was also documented that a bZIP (Basic Leucine Zipper) transcription factor HY5 (Long Hypocotyl5) acts as a molecular link between ethylene and ABA signaling [204,205]. HY5 is a part of the essential transcriptional cascade in ABA-modulating ethylene responses. ABA-induced HY5 binds to the G-box region of the *AtERF11* gene to activate its expression. The AtERF11, in turn, act as a repressor and represses the enzyme 1-aminocyclopropane-1-carboxylic acid synthase2 (ACS2) and ACC5 transcription, which catalyzes the rate-limiting step of ethylene biosynthesis [204]. It is suggested that HY5 acts as a negative regulator of ethylene response and a positive regulator of ABA response, but the exact mechanism is still unclear [206]. The *ACS2/5* genes have a negative effect on primary root development. ABA is the negative regulator of root hair growth, which is confirmed by the exogenous supply of ABA to the plants, but these results are not observed in the ABA-insensitive mutant. ABA controls the root hair growth by upregulation of DNA BINDING WITH ONE FINGER (DOF)-type transcriptional regulator OBF BINDING PROTEIN4 (OBP4), which binds to the promoter region of *ROOT HAIR DEFECTIVE6-LIKE2 (RSL2)* gene and inhibits its expression (Figure 5) [207]. Among the basic helix-loop-helix transcription factors that are known to regulate the root growth, RSL4 subsequently activates a group of genes that are specifically expressed in the root hair for growth.

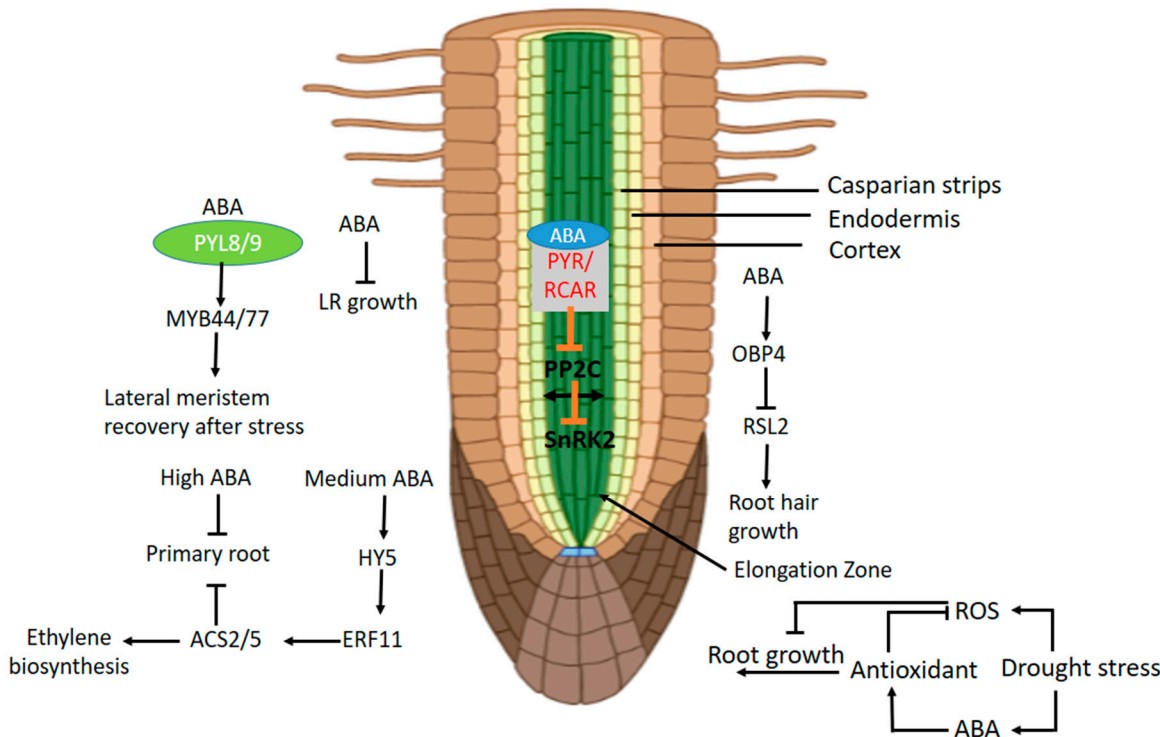

**Figure 5.** ABA modulates the root growth and architecture in response to changing environments. The intermediate level of ABA stimulates root growth to go deeper and seek water in drought conditions. The high-level ABA inhibits root growth and lateral root formation (the molecular mechanism was discussed in the text). ABA may act antagonistically or synergistically with other plant hormones and signals to fine-tune root architectures. The role of ABA in root growth depends on ABA concentration and may vary among plant species.

A high level of ABA reduces the primary root length in maize [208], rice [209,210], *Arabidopsis* [211,212], and Medicago [213,214]. In *Arabidopsis*, it is also demonstrated that ABA controls primary root length in a dose-dependent manner [215,216]. A high level of ABA inhibits the cell division in the QC and the proximal part of the root meristem through ethylene-dependent pathways. This can be carried out by suppressing the transcription of cell cycle genes, such as CYCLIND 3;1 (CYCD3;1) and CYCLIN DEPENDENT KINASE B1;1(CDKB1;1) [215], while a low level of ABA enhances root growth through ethylene-independent pathways using auxin signaling and a PIN2-dependent transport mechanism [189]. Furthermore, a low concentration of ABA enhances the QC's quiescence and inhibits the differentiation of stem cells and their daughter cells in the root meristems. The lateral roots of the plants are more sensitive to ABA than the primary roots. Therefore, low concentrations of ABA produce a stronger impact on lateral root elongation [211,215,217,218].

ROS play an important role in ABA signaling and negatively regulate auxin signaling to decrease the concentration of auxin in the roots [196]. ABA activates the NADPH oxidases ATrbohD and ATrbohF, which enhance the ROS synthesis and inhibit primary root elongation. This can be done through the downregulation of cell cycle-related genes [169,219], disruption of DNA replication, changes in the cellular redox balance, and damages to the cell's wall structure [220]. ROS also activate the plasma membrane $Ca^{2+}$ permeable channels to enhance the $Ca^{2+}$ level in the root, which inhibits the primary root growth through the activation of the CPK4 and CPK11 kinases [71,221]. However, further research is needed to elaborate on the underlying mechanisms.

### 8.3. Regulating the Role of ABA and GA in Seed Dormancy and Germination

Seed dormancy is an adaptive trait that keeps the seed viable, but restricts its germination even under favorable environmental conditions. That is why seed dormancy is considered to be a desirable trait, that prevents preharvest sprouting and ensures uniform germinations [222]. The mechanisms of seed dormancy and germination are under the control of two endogenous phytohormones: gibberellins (GA) and ABA. Both phytohormones work analogously: ABA plays a central role in the induction and maintenance of dormancy, and GA enhances germination [223,224].

Different treatments, including after-ripening, light, and cold stratification induce losses in seed dormancy. In cereal seeds, the dormancy loss prevails, with a change in the physiological state of the seed, which modifies the gene and protein expression, i.e., epigenetic alteration, and prompts an oxidative modification of gene transcripts and proteins [222,224]. Phytohormones GA and ABA are the major players in the regulation of dormancy and germination. Therefore, the change of the ABA/GA level and sensitivity is the main regulatory mechanism underlying the maintenance of dormancy and seed germination. It is well known that the balance of the ABA/GA level can be regulated by the expression of genes involved in the GA and ABA metabolism and signaling [225,226]. Environmental (light, temperature) and endogenous signaling factors (ROS) can also influence the balance between GA and ABA. Recent studies in genomics have identified the genes involved in GA and ABA metabolism and signaling pathways [226,227]. These studies have opened new paths to elucidate the molecular mechanism of GA and ABA, and their role in seed dormancy and germination. The ABA level in the seeds is regulated by biosynthesis and catabolism. Indeed, the biosynthesis of ABA is a several-step process, but 9-cis-epoxycarotenoid deoxygenase (NCED) is the rate-limiting factor. In catabolism, the *CYP707A* gene encodes for ABA 8′-hydrolase (ABA8′OH), whereby the hydroxylation of ABA at 8′ position occurs. Therefore, the expression of *NCED* and *CYP707A* genes has an important role in regulating the ABA level in dormant seeds [228,229].

The gibberellin signaling is initiated when the bioactive GA is perceived by its receptor, GIBBERELLIN INSENSITIVE DWAEF1 (GID1), which further induces the degradation of DELLA proteins (a negative regulator of GA signaling pathways). GA binds to GID1 and enhances the formation of the GA-GID1-DELLA complex, which, in turn, is associated with F-box proteins and degrades the DELLA protein through ubiquitin 26S proteasome pathways. The degradation of GA's negative regulator then mediates GA signaling (Figure 6) [230,231]. It was demonstrated that GA promotes seed germination by overcoming the mechanical barriers surrounding the embryos, and thus enhancing the growth potential of the embryo [232]. In wheat and barley seeds, after-ripening enhanced the TaGA20ox and TaGA3ox genes expression and increased the level of bioactive GA1 during imbibition, which helps in breaking dormancy [233,234]. In rice, the genetic studies identified the biosynthesis genes *OsGA20ox2* and *OsGA20 × 3* as candidates for controlling seed germination. Furthermore, the *OsGA20ox2* mutant led to a reduction of the GA level and enhanced the seed dormancy. The dynamic balance between GA and ABA metabolism can be modulated by their reciprocal regulation at the time of dormancy and germination. The imbibition of non-dormant barley seeds induced the expression of the GA biosynthesis gene (HvHA3ox2) and ABA catabolic gene (HvABA8′OH1), to increase GA and reduce ABA levels [235,236]. Previous studies identified the PP2C protein as a negative regulator of ABA signaling that enhances seed germination by modulating the GA metabolism and signaling genes. In rice, PP2C represses OsbZIP10 (Homolog of AB15) through dephosphorylation, and leads to the overexpression of OsPP2C51, to increase the expression and activity of $\alpha$-amylase and thereby promote seed germination [236]. PP2C mediates the GA and ABA responses between seed dormancy and germination, but more studies are needed to elucidate the role of PP2C in detail. In *Arabidopsis,* during the post-germination stage, ABI4 repressed the ABA catabolic genes (*CYP707A1* and *CYP707A2*) expression and promoted the expression of ABA biosynthetic (NCED6) and GA catabolic (GA2ox7) genes. However, further studies are required to determine the role of ABI4 in seed dormancy and germination [237]. In *Arabidopsis*, DELLA proteins regulated GA and ABA through crosstalk after interacting with AB13 and AB15. Furthermore, DELLA also interacts with a RING-HZ

zinc finger E3 ubiquitin ligase, XERICO, which regulates the ABA signaling. In a low concentration of GA, DELLA promotes the expression of XERICO, which enhances the transcription of AB15 and increases the ABA level to inhibit seed germination [238,239].

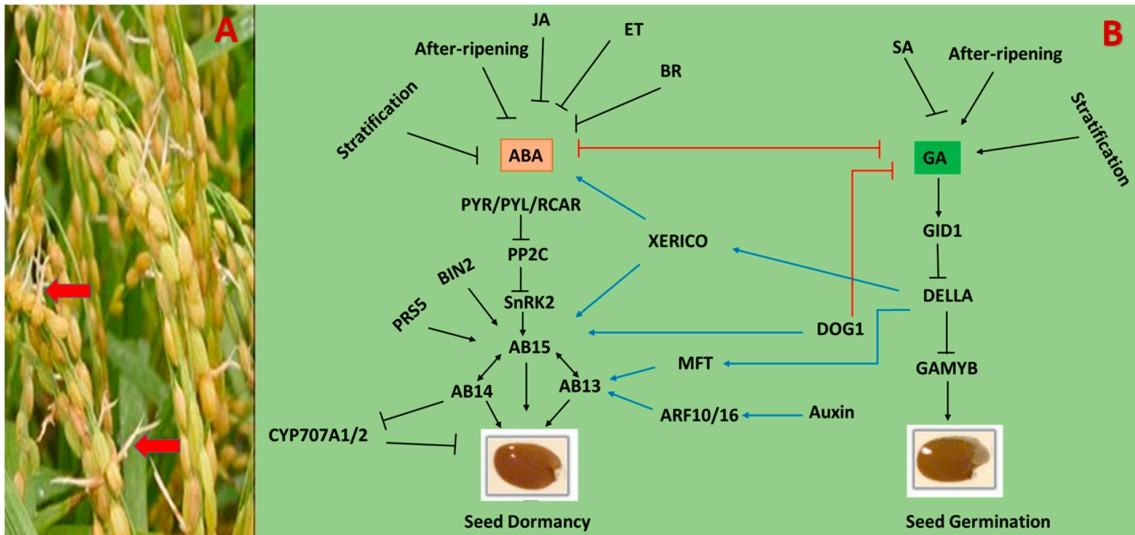

**Figure 6.** (**A**). Pre-harvest sprouting of the mature plants happens during the harvesting season, when plants face unfavorable environmental conditions of prolonged rainfall and high humidity. (**B**). The first phase of germination after ripening or stratification dormancy release is regulated by ABA and gibberellins (GA). Other phytohormones such as jasmonic acid (JA), brasenosteiod (BR), and ethylene inhibited the ABA accumulation and enhanced the dormancy release. ABA positively regulates seed dormancy by modulating the WRKY transcription factor, which inhibits GAMYB-mediated GA responses. GA repressor DELLA promotes the expression of XERICO, which increased the ABA level and AB15 activity and maintained the seed dormancy. MFT (MOTHER OF FT AND TF1 (MFT)), regulating the balance between the GA and ABA responses, involves AB15 and AB13, that act as the activator and repressor of MFT expression, and DELLA also activated the MFT expression. DELAY OF GERMINATION (DOG1) regulates ABA signaling by interacting with PP2C, modulating AB15, and interacting with AB13. AB15 is an important factor which is regulated at the transcriptional and post-transcriptional level; AB14 promotes its expression, while PKS5 and BIN2 phosphorylate it. See the caption in Figure 1 on ABA metabolism.

During the germination of cereal seeds, the transcriptional induction of α-amylase increases in order to degrade starch and transfer it to the germinating embryos. In the aleurone of barley seeds, the expression of α-amylase can be activated by GA-inducible *GAMYB* [240], while repressed by ABA-induced PKABA1 (Ser-Thr Kinase). In rice, MFT is observed to be the main factor in the ABA-enhanced inhibition of GA-inducible responses [241]. Recent studies have shown that DOG1 promotes seed dormancy by modulating the expression of AB15 and the interaction with AB13 [242]. However, further studies are needed to clear the role of MFT and DOG1 in the developmental switch of seed between dormancy and germination.

## 9. Conclusions and Future Perspectives

Food demand is rising day by day due to the global climatic change and growing population. Thus, food production must be increased to meet the required demand. Therefore, the recognition of abiotic stress tolerance in plant species has important implications to sustain food supplies that cannot be underestimated. ABA is a critical hormone and serves as a core regulator of various stresses, such as low temperature, salinity, and drought in the plant species. The developments in genome-wide and molecular genetic technology will develop our in-depth understanding of the ABA's role in stress tolerance. At the molecular level, various physiological processes (seed dormancy and germination,

stomatal closure, and change in gene expression) that are regulated by ABA have been described, but there is still space for further discoveries in the field. In fact, the discoveries related to the stress sensor will significantly refine our understanding of how a plant changes ABA accumulation in response to environmental stress. While some putative sensors related to cold, salinity, and osmotic stresses have been recognized, the linking mechanism between ABA response and environmental stresses is poorly understood. Comprehending the regulation of ABA requires a further exploration of upstream sensing and signaling events. Furthermore, further elaboration is also needed to answer the following unsolved questions; (a) How do plants respond to different stress stimuli? (b) How do plants react to stresses in different cells and tissue types? (c) How are responses synchronized and integrated in intercellular communication in different cells and tissues?

**Author Contributions:** Conception of the presented idea, S.A. and L.X.; development of the theory and computations, S.A. and K.H.; resources, S.A., A.I. and L.X.; writing—original draft, S.A. and K.H.; writing—review & editing, S.A., A.I. and L.X. All authors have read and agreed to the published version of the manuscript.

**Funding:** This work was supported by project 3180101344, National Natural Science Foundation of China.

**Conflicts of Interest:** The authors declare no conflict of interest.

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
