# Peer review of "Implications of Abscisic Acid in the Drought Stress Tolerance of Plants"

_agronomy, doi:10.3390/agronomy10091323_

Round 1

Reviewer 1 Report

Dear Authors,

The paper you have drafted is very interesting and relevant regarding the role of ABA in plants response to drought stress, from its biosynthesis to its mode of action. However, I found the article very difficult to read, for this reason I would like to ask you to please review the English language in order to enhance to quality of your work and facilitate its publication.

The comments I made are below:

Line 20: The site of its biosynthesis

Line 26: express this sentence better

Line 35: correct English verb tense

Line 43 to 45: express this sentence better

Line 54 to 57; not clear

Reduce the length of paragraph 2 by including essential information

Line 66 to 71: reformulate the sentence. It’s not clear

Line 71-73: fix this sentence. English language errors

Line 76: “which trigger downstream events, such as gene expression and protein phosphorylation”

Line 78-79: sentence not necessary

Line 86: “transgenic lines”

Line 90-91: this sentence is repetitive

Line 102: explain this sentence better or skip it. It’s not clear

Line 163: Do you mean “phytopathogenic fungi”??

Line 165: “Plants use the carotenoid...”

Figure 1: Correct the spelling of DPA and ABA-GE in the image and in the Image description for ABA-GE.

Line 324: Do you mean ABCG subfamily?

Line 372: “stomatal”

Line 500: No need to repeat the references twice

Thanking you in advance.

Best regards

Author Response

Reviewer 1

Dear Authors,

The paper you have drafted is very interesting and relevant regarding the role of ABA in plants response to drought stress, from its biosynthesis to its mode of action. However, I found the article very difficult to read, for this reason I would like to ask you to please review the English language in order to enhance to quality of your work and facilitate its publication.

Response: thank you so much for your comments and suggestion, we modified the whole MS thoroughly and made the extensive change to improve the quality of English, Errors in the tense, and grammar has been corrected.

The comments I made are below:

Line 20: The site of its biosynthesis

Response: the correction has been made in the MS

Line 26: express this sentence better

Response: the correction has been made in the MS; we rearranged this paragraph

Line 35: correct English verb tense

Response: the correction has been made in this sentence.

Line 43 to 45: express this sentence better

Response: the meaning of the sentence has been improved

Line 54 to 57; not clear

Response: the meaning of the sentence has been improved

Reduce the length of paragraph 2 by including essential information

Response: the correction has been made

Line 66 to 71: reformulate the sentence. It’s not clear

Response: the sentence has been modified

Line 71-73: fix this sentence. English language errors

Response: The English errors have been fixed

Line 76: “which trigger downstream events, such as gene expression and protein phosphorylation” Response: the correction has been made in the MS

Line 78-79: sentence not necessary

Response: the paragraph has been modified

Line 86: “transgenic lines”

Response: the correction has been made

Line 90-91: this sentence is repetitive

Response: This paragraph has been revised, and the sentence has been deleted in the
revised manuscript

Line 102: explain this sentence better or skip it. It’s not clear

Response: the correction has been made in the MS

Line 163: Do you mean “phytopathogenic fungi”??

Response: We apologize for the error found by the reviewer

Line 165: “Plants use the carotenoid...”

Response: the correction has been made in the MS

Figure 1: Correct the spelling of DPA and ABA-GE in the image and in the Image description for ABA-GE.

Response: thanks for the comment, the correction has been made in the figure 

Line 324: Do you mean ABCG subfamily?

Response: the correction has been made

Line 372: “stomatal”

Response: the correction has been made

Line 500: No need to repeat the references twice

Response: the correction has been made

Thanking you in advance.

Best regards

Reviewer 2 Report

I have some comments as following:

Line 67: The authors should introduce some researches related to the perception of drought stress signal before the topics of calcium. They can show the histidine kinases in response to abscisic acid, drought, and salt stress in Arabidopsis (Tran et al. 2007).

Tran et al. (2007) showed functional analysis of AHK1/ATHK1 and cytokinin receptor histidine kinases in response to abscisic acid, drought, and salt stress in Arabidopsis.

Tran LSP et al. (2007) Functional analysis of AHK1/ATHK1 and cytokinin receptor histidine kinases in response to abscisic acid, drought, and salt stress in Arabidopsis. PNAS 104 (51): 20623-20628; https://doi.org/10.1073/pnas.0706547105

Line 221: The authors should check the description of “Liu et al. 2014”.

Line 279; “Dehydration responsive” should be “dehydration-responsive”.

Line 324: I think that “ABACG” should be “ABCG”.

Others: In addition, Takahashi et al. (2018) reported a small peptide modulates stomatal control via abscisic acid in long-distance signalling. Recently, Kim et al. (2017) reported acetate-mediated novel survival strategy against drought in plants. Since these factors are involved in drought tolerance, please consider introducing them in this review.

Kim JM et al. (2017) Acetate-mediated novel survival strategy against drought in plants. Nature Plants, 10.1038/nplants.2017.97

Takahashi F et al. (2018) A small peptide modulates stomatal control via abscisic acid in long-distance signalling. Nature, 10.1038/s41586-018-0009-2

Author Response

Reviewer 2

I have some comments as following:

Line 67: The authors should introduce some researches related to the perception of drought stress signal before the topics of calcium. They can show the histidine kinases in response to abscisic acid, drought, and salt stress in Arabidopsis (Tran et al. 2007).

Tran et al. (2007) showed functional analysis of AHK1/ATHK1 and cytokinin receptor histidine kinases in response to abscisic acid, drought, and salt stress in Arabidopsis.

Tran LSP et al. (2007) Functional analysis of AHK1/ATHK1 and cytokinin receptor histidine kinases in response to abscisic acid, drought, and salt stress in Arabidopsis. PNAS 104 (51): 20623-20628; https://doi.org/10.1073/pnas.0706547105

Thank you so much for your comments and suggestions; we agree with these comments. We have read the whole manuscript and added a further description of the perception of drought stress signals and plasma membrane receptors such as AHK1/ATHK1, which further triggers the disruption of intracellular calcium level for activating phosphorylation cascades.

Line 221: The authors should check the description of “Liu et al. 2014”.

Response: the correction has been made in the MS.

Line 279; “Dehydration responsive” should be “dehydration-responsive”.

Response: the correction has been made in the MS.

Line 324: I think that “ABACG” should be “ABCG”.

Response: the correction has been made in the MS.

Others: In addition, Takahashi et al. (2018) reported a small peptide modulates stomatal control via abscisic acid in long-distance signalling. Recently, Kim et al. (2017) reported acetate-mediated novel survival strategy against drought in plants. Since these factors are involved in drought tolerance, please consider introducing them in this review.

Kim JM et al. (2017) Acetate-mediated novel survival strategy against drought in plants. Nature Plants, 10.1038/nplants.2017.97

Takahashi F et al. (2018) A small peptide modulates stomatal control via abscisic acid in long-distance signalling. Nature, 10.1038/s41586-018-0009-2

Response: thanks for sharing the paper links related to our study. The information related to these articles is inserted in the section of ‘’regulation of ABA biosynthesis through the abiotic stress’’.

Round 2

Reviewer 1 Report

Dear Authors,

Thank you for the corrections you have made. I just noted a few typos you can quickly correct .

All the best!

51: "regulator"
55:  " the plant species to adapt continuously to the changing"

62: "With the help"

76: "combat"

86: "to plant adapt to changes in the surrounding environment"

87: "Plants perceive the stimuli"

89: "long time or Time-lag?"

99:  "consist in a pair of Ca2+"

Author Response

Dear Authors,

Thank you for the corrections you have made. I just noted a few typos you can quickly correct.

All the best!

Response: thank you so much, we appreciate the time and effort that you have dedicated to providing your valuable feedback on our manuscript. We are grateful for your insightful comments on our paper.

51: "regulator"

Response: the correction has been made in the MS

55:  " the plant species to adapt continuously to the changing"

Response: corrected

62: "With the help"

Response: the correction has been made in the MS

76: "combat"

Response: the spelling mistake has been corrected

86: "to plant adapt to changes in the surrounding environment"

Response: the correction has been made in the MS

87: "Plants perceive the stimuli"

Response: corrected

89: "long time or Time-lag?"

Response: the correction has been made in the MS

99:  "consist in a pair of Ca2+"

Response: the correction has been made in the MS